# *BaZFP1,* a C2H2 Subfamily Gene in Desiccation-Tolerant Moss *Bryum argenteum*, Positively Regulates Growth and Development in *Arabidopsis* and Mosses

**DOI:** 10.3390/ijms232112894

**Published:** 2022-10-25

**Authors:** Ping Zhou, Xiujin Liu, Xiaoshuang Li, Daoyuan Zhang

**Affiliations:** 1State Key Laboratory of Desert and Oasis Ecology, Xinjiang Institute of Ecology and Geography, Chinese Academy of Sciences, Urumqi 830011, China; 2Xinjiang Key Lab of Conservation and Utilization of Plant Gene Resources, Urumqi 830011, China

**Keywords:** *Arabidopsis thaliana*, mosses, *BaZFP1*, plant growth, development

## Abstract

C2H2 zinc finger protein (C2H2-ZFP) plays an important role in regulating plant growth, development, and response to abiotic stress. To date, there have been no analyses of the C2H2-ZFP family in desiccation-tolerant moss. In this study, we identified 57 BaZFP transcripts across the *Bryum argenteum* (*B. argenteum*) transcriptome. The BaZFP proteins were phylogenetically divided into four groups (I–IV). Additionally, we studied the *BaZFP1* gene, which is a nuclear C2H2-ZFP and acts as a positive regulator of growth and development in both moss and *Arabidopsis thaliana*. The complete coding sequence of the *BaZFP1* gene was isolated from *B. argenteum* cDNA, which showed a high expression level in a dehydration–rehydration treatment process. The overexpression of the *BaZFP1* gene in the *Physcomitrium patens* and *B. argenteum* promoted differentiation and growth of gametophytes. Heterologous expression in *Arabidopsis* regulated the whole growth and development cycle. In addition, we quantitatively analyzed the genes related to growth and development in transgenic moss and *Arabidopsis*, including *HLS1*, *HY5*, *ANT*, *LFY*, *FT*, *EIN3*, *MUS*, *APB4*, *SEC6*, and *STM1*, and found that their expression levels changed significantly. This study may pave the way for substantial insights into the role of C2H2-ZFPs in plants as well as suggest appropriate candidate genes for crop breeding.

## 1. Introduction

Developmental processes in plants are triggered by a variety of external and internal factors. In higher plants, the transcriptional regulation of gene expression plays an important role in plant life. Transcription factors play important roles in the regulation of plant growth and development [1], as well as its response to the environment [2,3].

A large number of studies have shown that transcription factors are involved in plant growth and development. Zinc finger proteins (ZFPs) are a massive transcription factor family in the plant kingdom [4]. ZFP contains one to several zinc fingers, approximately 23 to 30 amino acids in length, and contains multiple cysteine and histidine residues. Based on the combination of cysteine and histidine residues, ZFPs can be classified into different types, such as C2H2, C2HC, C2HC5, C3HC4, CCCH, C4, C4HC3, C6, and C8 [5,6,7]. Among these types, Cys2/His2 zinc finger proteins (C2H2-ZFPs) are one of the largest and most diverse superfamilies [4,8]. The cysteine and histidine residues in the motif coordinately bind one or more zinc ions through hydrogen bonds, forming a stabilized finger-like structure that can interact with nucleic acid sequences to regulate target gene expression [8]. C2H2-ZFP refers to one of the most common C2H2-type (classical) zinc finger (ZF) motifs in ZFPs, which can be represented by the sequence X2-Cys-X2,4-Cys-X12-His-X3,4,5-His (X represents any amino acid residue, and the number represents the number of residues) [9,10]. Most C2H2-ZFPs in plants contain the highly conserved sequence QALGGH in their zinc finger domains; such proteins are referred to as Q-type zinc finger proteins. Unlike proteins containing the QALGGH sequence, a class of C2H2-ZFPs that do not contain any conserved motifs in the zinc finger region are known as C-type proteins [11,12]. The QALGGH sequence is thought to be a plant-specific motif. Since 1992, when the first plant C2H2-ZFP was cloned in the study of *Petunia petal* development, the investigation of C2H2-ZFP in flower development has been extensively studied [13]. Many studies have verified that C2H2-ZFPs participate in the transcriptional regulation of plant photomorphogenesis, seed germination, leaf growth, shoot development, gametogenesis, and floral organ development [14,15,16,17,18,19].

The moss *Physcomitrium patens* (previously named *Physcomitrella patens*) [20,21] exhibits a relatively simple developmental pattern, with alternating haploid gametophyte and diploid sporophyte generations [22]. The gametophyte consists of two distinct developmental stages, filamentous protonema, and adult gametophores, which are filamentous rhizoids and leafy shoots. In contrast to ferns and seed plants, the gametophyte is the dominant phase, and this generation comprises most of what is familiar to us as moss plants [23]. Many factors affect the differentiation process from protonema to gametophyte in *P. patens*.

*Bryum argenteum* Hedw. (*B. argenteum*) is an important component of the desert biological soil crusts in the Gurbantunggut Deserts of northwestern China [24,25]. As an extremely desiccation-tolerant moss, *B. argenteum* can survive under extremely dry air conditions (i.e., 0–30% RH or less than −162 MPa) [26]. In the previous work, we identified a putative C2H2-ZFP of *B. argenteum* (BaZFP1), which showed 20-fold induced gene expression in the late rehydration stage [27]. In this paper, we cloned the gene *BaZFP1* and focused on the function of this gene. Our data show that overexpression of the *BaZFP1* gene in mosses and *Arabidopsis* regulates both growth and development. Our finding provides both essential understandings of the function and characteristics of BaZFPs along with candidate genes for crop breeding.

## 2. Results 

### 2.1. Phylogenetic Analysis and Classification of BaZFP Proteins

Integrating the validations of online programs of Swiss-Prot, NCBI CDD, and SMART manual inspection in this study with the previous reports [28,29,30], we confirmed 156 C2H2-ZFP genes in the TAIR 10 genomes of *A. thaliana*. To determine the phylogenetic relationship between all 57 BaZFPs, a maximum likelihood (ML) tree was constructed with 57 BaZFPs and 156 AtBaZFPs (Figure 1). The ML phylogenetic tree of BaZFP full-length proteins was clustered into four groups (I, II, III, and IV). Group II diverged into four clades, which were clade IIa (25 members), IIb (24 members), IIc (27 members), and IId (23 members). In addition, group IV diverged into three clades, which were clade Iva (29 members), IVb (22 members), and IVc (23 members). Group II-a, II-b, II-d, and II-e had the zinc finger motif of CX2CX(4~19)HX(3~7)H. Group IV-a, IV-b, and IV-d, and II-e had the zinc finger motif of CX4CX(11~20)HX(2~5)H.

### 2.2. Identification and Physicochemical Properties of BaZFPs

A total of 57 BaZFP transcripts were obtained from the *B. argenteum* transcriptome after screening domains and removing redundant transcripts (Appendix A). In addition, the quantity of amino acids, molecular weight (MW), pI, and subcellular localization of these BaZFP transcripts were analyzed. These 57 putative BaZFP transcripts ranged from 67 (TR118432|c0_g1_i1) to 1687 amino acids (aa) (TR129315|c0_g1_i2) in length. The MWs of these transcripts varied from 7285.3 Kd in TR118432|c0_g1_i1 to 192505.55 Kd in TR129315|c0_g1_i2, and their predicted pI ranged from 4.45 (TR81818|c0_g1_i1) to 9.32 (TR90564|c0_g1_i1). Moreover, these transcripts were nuclear-localized according to the results of subcellular localization analysis, except TR108277|c1_g1_i1, TR105106|c0_g1_i1, TR121284|c0_g1_i1, and TR132382|c0_g1_i1. 

### 2.3. Conserved Motif Composition Analysis of C2H2 Proteins

The gene structures and motif characteristics of BaZFP transcripts were analyzed. MEME was used to search for coding motifs in all of the BaZFP transcripts sequences (Figure 2). Motif 1 is not contained in TR99120c0_g1_i1, TR85279c0_g1_i1, TR74415c0_g1_i1, TR55015c0_g1_i1, TR73396c0_g1_i1, TR99120c0_g1_i1, TR132084c0_g1_i1, and TR84852c0_g1_i1, but is in other BaZFP transcripts, which suggests that motif 1 could be a conserved motif among C2H2-ZFPs in *B. argenteum*. Four patterns are characterized by zinc finger (ZF) that differ in their spacing between the two invariant zinc-coordinating histidine residues by different numbers of amino acid residues, including A (CX1CX12HX (3-8) H), B1(CX2CX (4-19) HX (3-1)), B2 (CX2CX12HX (3~5), C (CX4CX (11-20), and HX (2-5) H). In addition, 49 of the 57 transcripts contained at least one type of B1 ZF, 24 of the 57 transcripts contained at least one type of C ZF, 6 transcripts contained one type of A ZF, and 9 transcripts contained at least one type of B2 ZF. These findings suggest that BaZFP transcripts with various motifs and C2H2-ZFs are possibly associated with corresponding functions and might function divergently based on phylogenetic analysis.

### 2.4. Bioinformatics Analyses of the BaZFP1 Gene and Its Encoded Protein

In this study, a ZFP gene (TR33924|c0_g1_i1) was cloned and named *BaZFP1*. The gene *BaZFP1* contained a 723 bp open reading frame (ORF) encoding a 231-amino-acid (aa) putative polypeptide. Based on the amino acid sequence alignments, *BaZFP1* contains two B1s and two type C ZFs, and the ZF domains do not contain a highly conserved QALGGH sequence (C-type zinc finger protein) (Appendix A). Subsequently, a phylogenetic tree was constructed by the neighbour-joining method based on amino acid sequences from algae, monocots, and dicots (Appendix A). Phylogenetic analysis of *BaZFP1* homologous genes in different plants showed that *BaZFP1* was more closely related to homologous proteins in *Marchantia polymorpha* (Mapoly0068s0079).

### 2.5. Transactivation Activity and Subcellular Localization Analysis of BaZFP1

The construct pGBKT7-*BaZFP1* was transformed into the yeast strain Y2H Gold. The yeast cells containing a pGBKT7 empty vector were used as a negative control, and the cells cotransformed with pGBKT7-53 were used as a positive control. As shown in Figure 3a, the transformants containing pGBKT7-*BaZFP1* and the positive control grew well not only on SD/-Trp medium but also on SD/-Trp/-His/X-α-Gal medium and displayed α-galactosidase activity. In contrast, yeast cells harbouring a negative control could only grow on SD/-Trp medium and did not exhibit α-galactosidase activity. These results suggest that *BaZFP1* has transcriptional activation activity in yeast. Full-length and truncated versions of *BaZFP1* were also fused into the pGBKT7 vector, and the transformants were screened on SD/-Trp/-His/X-α-Gal. Deletion of the domain from the position of 1 to 121 aa did not affect the activation, and the transactivation activity position was determined to be 200–241 aa. These results confirm that *BaZFP1* is a transcription activator and that the C-terminal has transactivation activity.

Subcellular localization of *BaZFP1* was examined by monitoring GFP fluorescence in onion epidermis cells transformed with either the fusion construct (*BaZFP1*-GFP) or the control (GFP). When onion cells were transformed with the GFP plasmid, green fluorescence signals were observed in the entire cell region. In contrast, fluorescence was exclusively detected in the nucleus of cells transformed with the fusion plasmid, indicating that *BaZFP1* was exclusively localized in the nucleus (Figure 3b).

### 2.6. BaZFP1 Overexpression Promotes Growth and Gametophyte Differentiation in B. argenteum and P. patens

The overexpression vector *BaZFP1* was transferred into the desiccation-tolerant moss *B. argenteum*, and obtained transgenic and wild-type *B. argenteum* were cultured on KNOP medium (Figure 4). After growing for 10 days, it was found that the protonema area of transgenic *B. argenteum* was significantly larger than that of the WT. Then, the number of regenerated gametophytes after 20 and 30 days of growth was counted. The results show that the regenerated shoot number and length of the transgenic *B. argenteum* were consistently larger than those in the wild type. In the late stage of gametophyte growth (after 50 days of growth), the gametophyte length of transgenic moss was also longer than that of the WT.

Similarly, we also transferred the overexpression vector into the model moss *P. patens* to observe whether *BaZFP1* affected the growth and development of *P. patens*. Consistent with the observation method of *B. argenteum*, the protonema of moss *P. patens* was transferred to KNOP medium and grown for 10, 20, and 30 days, and adult gametophyte (50 days of growth) (Figure 5). After 10 days of growth, the protonema area of transgenic moss was larger than that of the wild type. After 20 days of growth, the number of newly born gametophytes of transgenic moss was significantly higher than that of the wild type. At this stage, the transgenic lines were mainly in the meristematic gametophyte stage, while the WT was still mainly in the growth protonema stage. After 30 days of growth, the results show that the fresh weight of transgenic moss was significantly heavier than that of the wild type. At this stage, the transgenic lines were mainly growth gametophytes, while the wild type was mainly undergoing gametophyte differentiation. In the gametophyte stage, after 50 days of growth, the number of gametophyte leaves of transgenic lines was greater than that of the wild type, and the fresh weight was greater than that of the WT (Figure 4c).

### 2.7. BaZFP1 Overexpression Regulates Arabidopsis Growth and Development

We further tested the performance of *BaZFP1* in *Arabidopsis*. Col-0- and *BaZFP1*-overexpressing lines grown vertically on Murashige and Skoog (MS) solid medium for 3 days in the dark after germination were used to evaluate the effect of *BaZFP1* on the hook angle. Compared with the WT, the transgenic lines almost did not form a hook (Figure 6), which indicates that overexpression of *BaZFP1* affects the formation of hooks in *Arabidopsis* under dark conditions. We further tested the roles of *BaZFP1* in transgenic line growth under normal conditions. After 5 days of growth, the transgenic hypocotyl was longer than the WT plants. In addition, after 12 days of growth in soil, the fresh weight and leaf size of the transgenic lines were significantly larger than those of the WT. In the flowering stage, after 25 days of growth, obvious bolting appeared in WT plants, while the transgenic lines were still maintained in the vegetative growth stage. The beginning of the bolting time of transgenic lines was about 15 days later compared to the WT. Subsequently, after 50 days of growth, the WT had finished fruiting and showed senescence, while the transgenic lines still kept flowering and showed a much higher plant height than the WT. After 10 more days, all the transgenic lines finished fruiting. Finally, the seeds of transgenic lines were golden in colour and fuller than those of the WT and showed significantly higher thousand-seed weight. Therefore, these results suggest that the *BaZFP1* gene can significantly regulate growth and development in *Arabidopsis*. This evidence further shows that the *BaZFP1* gene can not only regulate the growth and development of moss but also play the same role in *Arabidopsis*.

### 2.8. Quantitative Analysis of Growth- or Development-Related Gene Expression Profiling

To investigate the possible molecular mechanisms of *BaZFP1* function in plant growth or development, the qRT-PCR approach was used to identify the genes with altered expression levels in the *BaZFP1* transgenic lines. We identified genes related to growth and differentiation in *P. patens and B. argenteum*, including *APB4*, *SEC6*, *STM1*, and *DCL4*, to analyze the expression patterns in transgenic moss lines (Figure 7). In the two mosses, the expression levels of *APB4* and *SEC6* increased significantly. Specifically, the expression of the *PpAPB4* gene was upregulated by at least 8-fold, while several other genes were increased at least 1.5-fold. However, the expression levels of *PpSTM1* and *PpDCL4* decreased by approximately 70% of the WT, and the expression levels of *BaSTM1* and *BaDCL4* decreased to approximately 40% and 70% of the WT. The quantitative results show that the expression patterns of these genes were similar in the mosses *P. patens* and *B. argenteum.*

Quantitative RT-PCR revealed that the expression levels of the growth-related genes, including *BaZFP1* apical-hook-formation-related genes (*HLS1*, *SAUR17*, *EIN3)*, hypocotyl-growth-related genes (*HY5*, *GAI*, and *PIF4*), vegetative-growth-related genes (*ANT*, *GRF3*, and *AN3*), and flowering-related genes (*FLY*, *FLC*, and *FT*), varied at different growth and development stages in *Arabdopsis* lines (Figure 8). In the curved-hook formation and expansion stage, the expression of *HLS1* and *SAUR17* was downregulated, with approximately 10% and 70% of the WT, respectively, and the expression of the *EIN3* gene was upregulated by at least 2-fold. After 5 days of normal growth, hypocotyl elongation was mainly observed. At this stage, the expression of the *HY5* and *PIF4* genes was downregulated to approximately 50% of the WT, while the expression of the *GAI* gene was upregulated by approximately 3-fold. In the vegetative growth stage (12 days), the expression levels of the *ANT, AN3*, and *GRF3* genes increased approximately 3-fold, 5-fold, and 3-fold, respectively. In the flowering stage (25 days), the expression levels of *LFY* and *FT* decreased to approximately 50% and 60% of the WT, respectively, while the expression of the *FLC* gene increased 10-fold. These results show that *BaZFP1* regulates key genes at different developmental stages in plants to affect plant development and growth, in moss and *Arabidopsis*.

## 3. Discussion

### 3.1. Conserved Motif Composition and Classification of BaZFPs

C2H-ZFPs are involved in the regulation of plant growth and development [31,32]. In this research, a total of 57 C2H2-ZFP (BaZFP) transcripts were identified through the *B. argenteum* transcriptome database. Then, the phylogenetic tree, physical and chemical properties, gene structure, conserved motifs, and ZF type were investigated. The phylogenetic analysis demonstrated that BaZFPs could be classified into four groups, among which group II was further divided into four subgroups, which was mainly concentrated in subgroup IId and the subgroup IVb. Moreover, some of the *Arabidopsis* C2H2-ZFP genes and the BaZFPs were clustered into the same clade, suggesting that they may have similar functions under abiotic stress and a close evolutionary relationship. Plant C2H2-ZFPs have unique structural features: the conserved, unique QALGGH (Q-type C2H2) motif and the long variable spacers between adjacent ZF domains [33]. However, unlike most plant C2H2-ZFPs containing the QALGGH domain that is involved in DNA binding [34], only one transcript in the BaZFP family contains a conserved domain. In the C2H2-ZFPs, it can be interactions by the leucine-rich box (L-box) or directly to target downstream genes, when no conserved QALGGH domain is present [35,36]. Additionally, most plants contain one to two ZF structures, but 29 transcripts in the BaZFPs contained more than three ZF structures; this may be due to some changes in evolution and deserves further investigation. In this study, *BaZFP1* from *B. argenteum* was identified and characterized, and we demonstrated that the *BaZFP1*-deduced polypeptide sequence is a member of the C2H2 family and most similar to deduced polypeptide sequences from *M. polymorpha* (Mapoly0068s0079). In plants, C2H2-ZFPs share a similar structure. *BaZFP1* is located in the nucleus and shows transcriptional activation activity, like most other plant C2H2-ZFPs [37]. In addition, we found that *BaZFP1* had four adjacent C2H2 finger structures. Since most of the C2H2-type TFs of plants harbour two ZFs, which seem to be sufficient for DNA binding, the extra ZFs may contribute to enhancing the specificity of the transcription factor to the promoters of the target genes [38]. The four adjacent ZF structures and their possible DNA binding mode and potential function need more study. 

### 3.2. Overexpression of BaZFP1 Regulates the Growth and Development of B. argenteum and P. patens

We found that *BaZFP1* transgenic moss had the same trend between *B. argenteum* and *P. patens*, increasing the differentiation and growth of gametophytes. Previous studies have found that the BABY BOOM (APB) is necessary for the formation of gametophore apical cells from protonema cells in *P. patens* [39], and expression of the *APB4* gene can increase the number of gametophytes. In this study, the expression level of the *APB4* gene increased significantly in two transgenic mosses. The *PpSEC6* gene plays an important role in gametophyte development [40]. Our results show that the expression of the *SEC6* gene was significantly induced in the two transgenic mosses. DICER-LIKE A (*PpDCL*) is an essential miRNA-directed cleavage of the target [41,42]. In *Arabidopsis dcl4*, mutants can accelerate changes in juvenile adult trophic stages [43]. In this study, the *DCL4* gene was suppressed, which likely accelerated the differentiation of the moss from the protofilament to the gametophytes. In addition, the STEMIN1 (*STM1*) genes facilitate cell reprogramming and the forming of new individuals. However, we found that the expression level of the *STM1* gene in the transgenic lines in the two mosses was significantly lower than that in the WT, which is worthy of further research. 

### 3.3. Heterologous Expression of BaZFP1 Regulates the Growth and Reproduction Process in Arabidopsis

In *Arabidopsis*, the formation of apical hook development and maintenance is regulated by multiple factors, including decreased (HOOKLESS1) *HLS1*-expression-inhibited apical hook formation [44,45], *EIN3* combining with *HLS1* to affect the formation [46], and *SAUR17* being closely related to the formation and relaxation [47]. Our results show that the expression level of both *HLS1* and *SAUR17* decreased greatly in transgenic lines, which is consistent with previous research, and shows that the *BaZFP1* gene regulates hook formation to decrease the expression of *HLS1* and *SAUR17* genes. 

During the hypocotyl elongation stage in *Arabidopsis*, *HY5* (long hypocotyl 5) and transcription factor *PIF4* (phytochrome-interacting factor) inhibit hypocotyl elongation [48], and *GAI* (GA-INSENSITIVE) encoding DELLA proteins promotes elongation growth [49]. Our quantitative results show that both *HY5* and *PIF4* gene expression levels were significantly lower in the transgenic lines, and the *GAI* gene was significantly increased. The results indicate that the *BaZFP1* gene may regulate transgenic *Arabidopsis* hypocotyl elongation by inhibiting *HY5* and *PIF4* genes and increasing *GAI* genes; however, the specific pathway by which the *BaZFP1* gene regulates deserves further study.

For the vegetative growth stage, The ANGUSTIFOLIA3 (*AN3*) can increase vegetative biomass during the development of *Arabidopsis* leaves [50]. In addition, the *GRF3* gene is involved in many developmental processes, including leaf development, stem elongation, and flower organ [51], AINTEGUMENTA (*ANT*) is necessary for control plant growth and floral organogenesis [52,53,54]. In our study, the *AN3* and *GRF3, ANT* gene expression levels were greatly increased in the transgenic *Arabidopsis* lines, which may explain the presence of significantly more vegetative biomass in transgenic lines compared with the WT. 

At the flowering stage, the flowering time of transgenic lines was approximately 15 days later than that of the WT. Therefore, we quantified flowering-time-related genes in *Arabidopsis*. FLOWERING LOCUS T (*FT*) promotes the transition to reproductive development and flowering [55,56]. Quantitative results reveal a significant reduction in the expression levels of the *FT* gene in the transgenic lines. *FLOWERING LOCUS C* (*FLC*) functions as a key flowering repressor in *Arabidopsis* [57,58]. Via overexpressing the *BaZFP1* gene, *FLC* genes were upregulated over 10-fold. The gene *LEAFY* (*LFY*) of *Arabidopsis* plays a controlling role in several aspects of flower development [59,60]. In the *BaZFP1*-overexpressing lines, the expression level of *LFY* was lower than that in the WT. Delayed flowering time is due to a prolonged plant vegetative growth time. Longer vegetative growth periods or a delayed flowering time can help plants absorb more nutrients to increase plant height and seed weight. Using the properties that increase biomass growth and delay flowering time, the *BaZFP1* can be transferred into grasses such as alfalfa and some green vegetables. There are relatively few studies on C2H2 transcription factors in moss, and the mechanism of the strong regulation of plant growth and development derived from the desiccation-tolerant moss *BaZFP1* is worthy of further study.

## 4. Methods and Materials

### 4.1. DNA/Protein Sequence and Phylogenetic Analyses

The transcriptome database of *B. argenteum*, which was published with the BioProject accession number PRJNA327617, was used. All these plant C2H2 protein sequences were queries to search the C2H2 family through local protein blast (BLASTP) against the transcriptome sequence database [18], with an E-value threshold of 0.00001. To obtain the 166 exact C2H2-ZFPs in *A. thaliana*, the *A. thaliana* protein sequences were downloaded from TAIR (https://www.arabidopsis.org/, accessed on 1 August 2022). In addition, redundant protein sequences were removed using the CD-Hit website with the default parameters [19]. The remaining C2H2 sequences were confirmed by Pfam (http://pfam.xfam.org/, accessed on 27 March 2022) and SMART (http://smart.embl.de/, accessed on 2 March 2022) [20,21]. Phylogenetic analysis was performed with the maximum likelihood (ML) method, with 1000 bootstrap replicates in MEGA 11.0. Additionally, the phylogenetic tree was modified by Figtree v1.4.4. Subsequently, the theoretical isoelectric point (pI) of these putative BaZFPs was determined [22]. Finally, we used WoLF-PSORT to predict the subcellular localization of BaZFPs (https://wolfpsort.hgc.jp/, accessed on 3 March 2022) [23].

Sequence alignment analysis of *BaZFP1* was performed using DNAMAN (version 6.0.3.99, Lynnon Biosoft, San Ramon, CA, USA). We used the Phytozome database (https://phytozome-next.jgi.doe.gov/ accessed on 12 December 2021) to find homologues of *BaZFP1*. A molecular phylogeny was constructed using MEGA (version 11.0, Mega Limited, Auckland, New Zealand) with the neighbour-joining method and 1000 bootstrap replicates [61], utilizing the monocot, fern, and bryophyte C2H2 sequences obtained from public resources. All C2H2 gene domain predictions were conducted with the MEME tool (https://meme-suite.org/meme/tools/meme, accessed on 25 December 2021).

### 4.2. Vector Construction and Plant Transformation

According to the coding sequence, the 35S:*BaZFP1* overexpression vector was constructed. The whole open reading frame (ORF) of *BaZFP1* was fused into the SmaI site of the plant expression vector 35S in front of the 35S promoter using the infusion PCR cloning method (TaKaRa, Beijing, China). The primer information is listed in Appendix A. Similarly, *SmaI* was used as the digestion site to construct the PBI121-GFP vector using an in-fusion PCR cloning system (TaKaRa, Beijing, China).

The 35S:*BaZFP1* overexpression vector was transformed into *B. argenteum* and *P. patens* separately for overexpression studies. The transformation method is a stable genetic transformation method mediated by *Agrobacterium tumefaciens* established in the early stage of the establishment (Invention patent No.: 202110185973,7). In addition, the 35S:*BaZFP1* overexpression vector was introduced into *Arabidopsis* transformation, followed by the floral dip method [62]. The *BaZFP1*-GFP and empty vector were transformed into onion epidermis and used for subcellular location studies. *Arabidopsis* T1 seeds were harvested, dried at 25 °C, and germinated on MS medium containing 50 μg mL^−1^ kanamycin to select transgenic seedlings. Seeds obtained from the primary transgenic lines were germinated on antibiotic-containing medium, and PCR analyses were performed on resistant plants. The T3 seeds were used as materials for all subsequent experiments.

### 4.3. Transactivation Activity in Yeast

For the transactivation assay, the full-length and assorted truncated fragments of *BaZFP1* were subcloned into the GAL4 DNA-binding domain of the pGBKT7 vector using the In-Fusion HD Cloning kit (TaKaRa, Beijing, China) to produce the pGBKT7-*BaZFP1* plasmid. Y2H Gold yeast cells harbouring pGBKT7-*BaZFP1* were streaked on SD/-Trp media in plates to observe yeast growth at 30 °C for 3–4 days. A staining assay was performed by adding 20 mgL^−1^ 5-bromo-4-chloro-3-indolyl-α-D-galactopyranoside (X-α-gal) to the SD/-Trp/-His medium.

### 4.4. Subcellular Localization

The entire coding sequence (CDS) of *BaZFP1* without the termination codon was amplified using primers with SmaI sites. The fragment was ligated into the SmaI sites of pBI121–GFP by infusion to produce the *BaZFP1*–GFP fusion protein driven by the CaMV35S promoter. The resulting plasmids were confirmed by sequencing and further used for subcellular localization. Transient expression assays in onion epidermal cells were conducted using *Agrobacterium tumefaciens* EHA105 cells. The localization of the fusion protein was observed using a confocal microscope (Zeiss, Jena, Germany). Images of GFP fluorescence, DAPI, and the bright field from the onion epidermal cell expression assay were acquired simultaneously and merged.

### 4.5. Plant Materials and Growth Conditions

*B. argenteum* and *P. patens* gametophytes were cultured on solid KNOP medium [63] at pH 5.8 in closed Petri dishes under controlled conditions: 22 °C, 16:8 h light: dark photoperiod, RH = 60% with a light intensity of 100 μmol, as described previously [64,65].

The *Arabidopsis* Columbia-0 (Col-0) ecotype was used as the wild type, and transgenic *Arabidopsis* plants were produced. Seeds of *Arabidopsis thaliana* were germinated on solid MS medium after surface sterilization [66]. Seedlings were transferred to 5 cm-diameter pots at the 4–6 leaf stage containing autoclaved peat substrate (Pindstrup, Mosebrug, Ryomgård, Denmark) and grown under controlled conditions at 22 ± 2 °C with a 14 h light/10 h dark cycle.

### 4.6. QRT-PCR Analysis to Analyze the Expression Pattern of Growth- or Development-Related Genes

Mosses grown for 20 days were collected for subsequent quantitative materials. *Arabidopsis* was collected on 3, 5, 12, and 25 days and used as a subsequent quantitative material. Total RNA was extracted from different samples with an HP Plant RNA Kit (Omega, Beijing, China) and reverse-transcribed to cDNA using PrimeScript™ RT Master Mix reverse transcriptase (TaKaRa, Beijing, China). Quantitative real-time PCR analysis was performed with TB Green^®^ Premix Ex Taq™ II (TaKaRa, Beijing, China). The relative expression level was calculated with the comparative 2^−Δ^^Δ^^CT^ method [67]. The primers used in this study are listed in Appendix A. The wild-type gene actin was used as the internal control. Each experiment was carried out for three biological replicates, each of which corresponded to three technological repeats of separate experiments.

### 4.7. Statistical Analysis

In mosses, 20 individuals were taken as a group when counting fresh weight. The fresh weight of *Arabidopsis* seeds was 1000-grain weight. Values are expressed as the mean ± standard deviation (SD). One-way ANOVA with Tukey’s multiple comparison test was conducted using IBM SPSS Statistics. For all statistical tests, the level of significance was set at * *p* < 0.05 and ** *p* < 0.01.

## 5. Conclusions

We identified stress-induced *BaZFP1* from the desiccation-tolerant moss *B. argenteum* and demonstrated that *BaZFP1* not only improved growth development but also promoted the differentiation process of both *P. Patent* and *B. argenteum*. *BaZFP1* regulates plant growth and development by regulating key growth- or development-related genes in plants (Figure 9). This study revealed that the C2H2 transcription factor from bryophytes plays a key role in regulating the growth and development of bryophytes and dicotyledons. Overexpression of the *BaZFP1* in some grasses or green vegetables can increase biomass and delay flowering time and greatly increase yield. It has found valuable candidates for crop breeding.

## Figures and Tables

**Figure 1 ijms-23-12894-f001:**
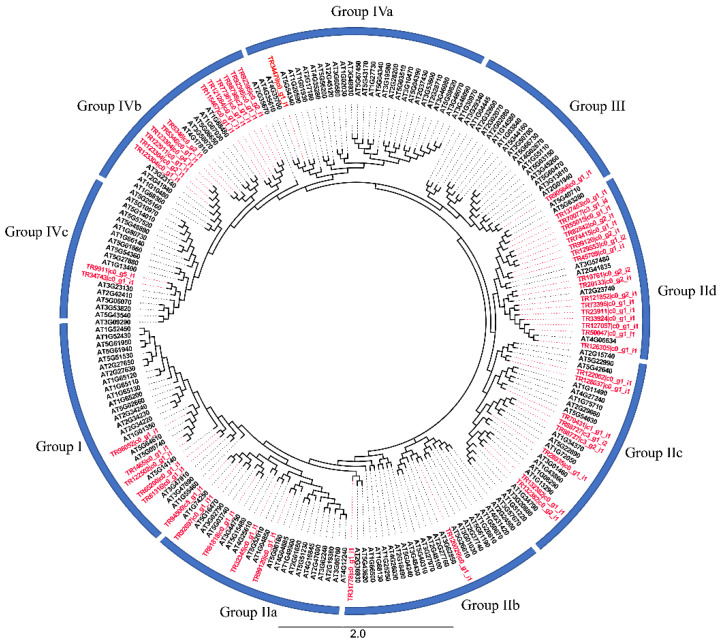
Phylogenetic analysis of the BaZFP proteins among *B. argenteum* (red) and *A. thaliana* (black). The phylogenetic tree was constructed using the maximum likelihood (ML) tree method.

**Figure 2 ijms-23-12894-f002:**
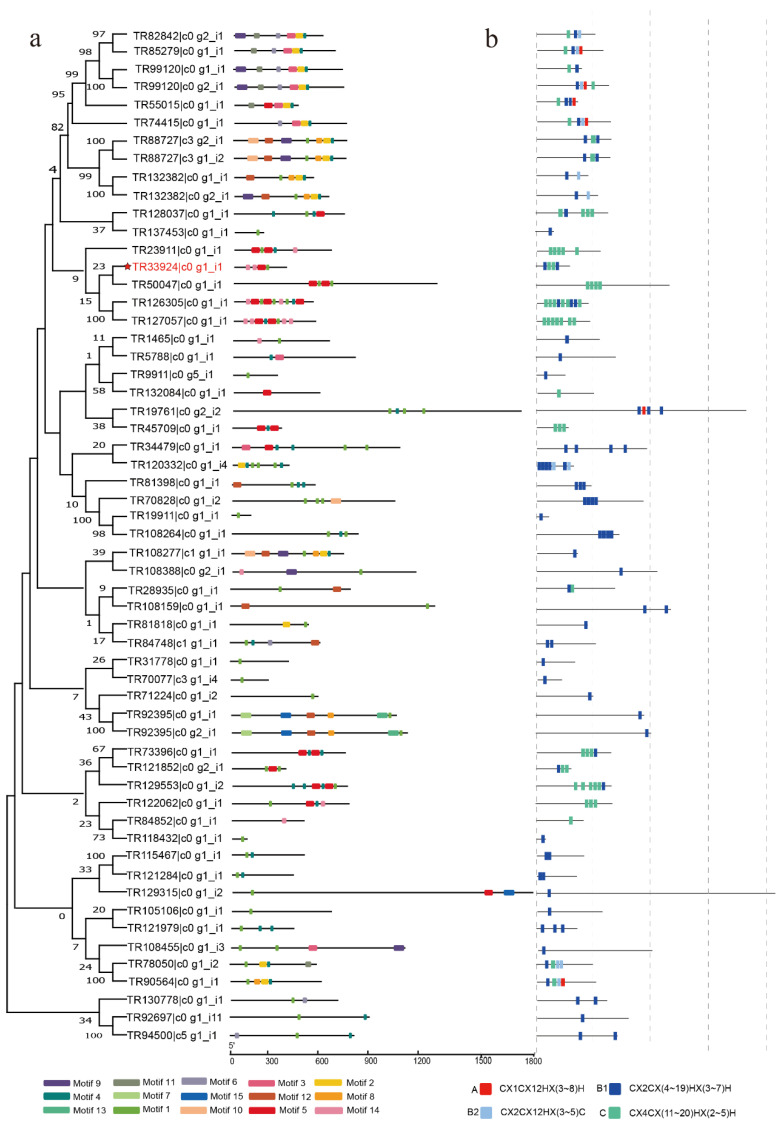
Motif and gene structures of C2H2 protein members in *B. argenteum*: (**a**) structure of conserved motifs; (**b**) different ZF types.

**Figure 3 ijms-23-12894-f003:**
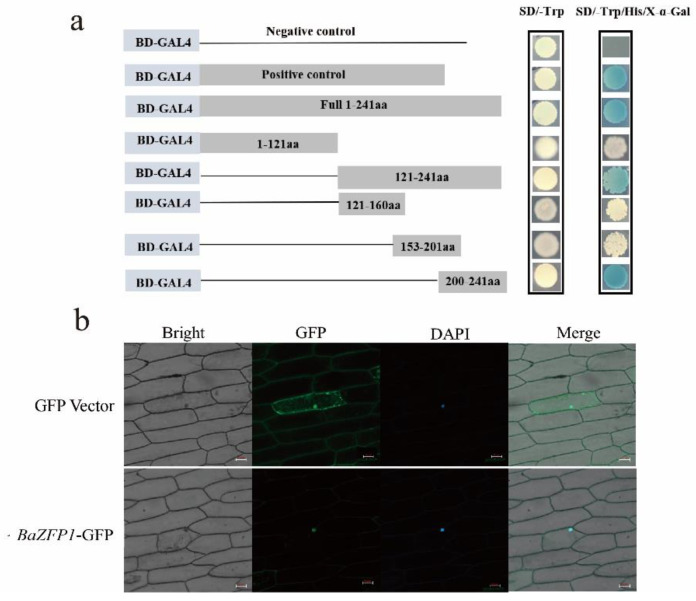
Transactivation activity and subcellular localization analyses of *BaZFP1*. (**a**) Transactivation analyses of *BaZFP1*. The fusion proteins of the GAL4 DNA-binding domain and *BaZFP1* were expressed in the yeast strain Y2H Gold. Truncated *BaZFP1* was fused with GAL4 BD, the vector pGBKT7 was used as the negative control, PGBKT7-53 was used as the positive control, and full-length *BaZFP1* was fused with the GAL4 BD domain. (**b**) Subcellular localization analysis of *BaZFP1*. Subcellular localization of *BaZFP1* in onion epidermis cells. The onion epidermis was transiently infiltrated with *A. tumefaciens* EHA105 expressing GFP vector or *BaZFP1*-GFP. Confocal images of the peeled epidermis were captured 72 h after inoculation at 488 nm by an LSM-800 laser scanning confocal microscope. Scale bars are 50 μm.

**Figure 4 ijms-23-12894-f004:**
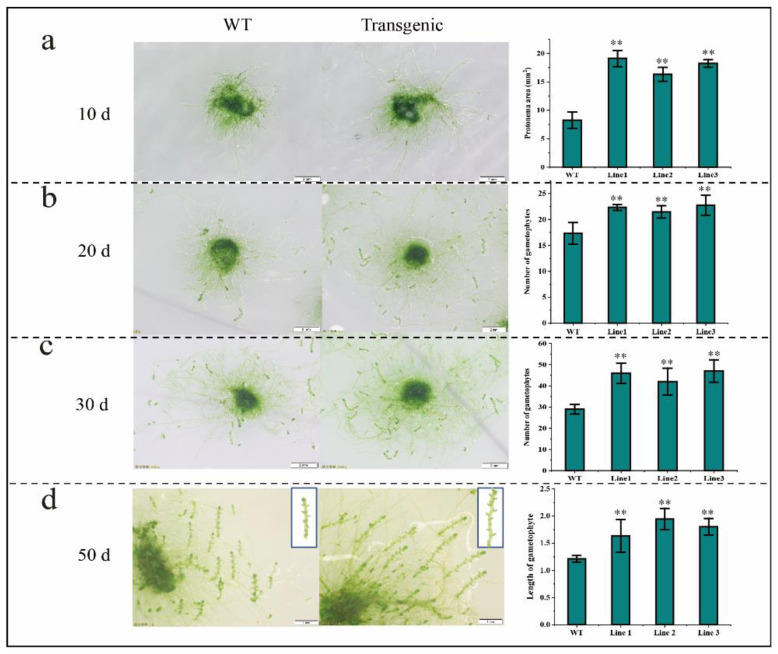
Phenotype of wild-type and *BaZFP1*-overexpressing lines of phenotype and data statistics in *B. argenteum*. (**a**) The phenotype of *B. argenteum* protonema transferred to KNOP medium for growth for 10 days, protonema area statistics; (**b**) the phenotype growth for 20 days, number of gametophytes statistics; (**c**) the phenotype growth for 30 days, length of gametophyte statistics; (**d**) the phenotype growth for 50 days, length statistics of gametophytes. Three independent biological replicates were performed, and vertical bars refer to ±SD (n = 3). Asterisks indicate significant differences between WT and transgenic lines (** *p* < 0.01).

**Figure 5 ijms-23-12894-f005:**
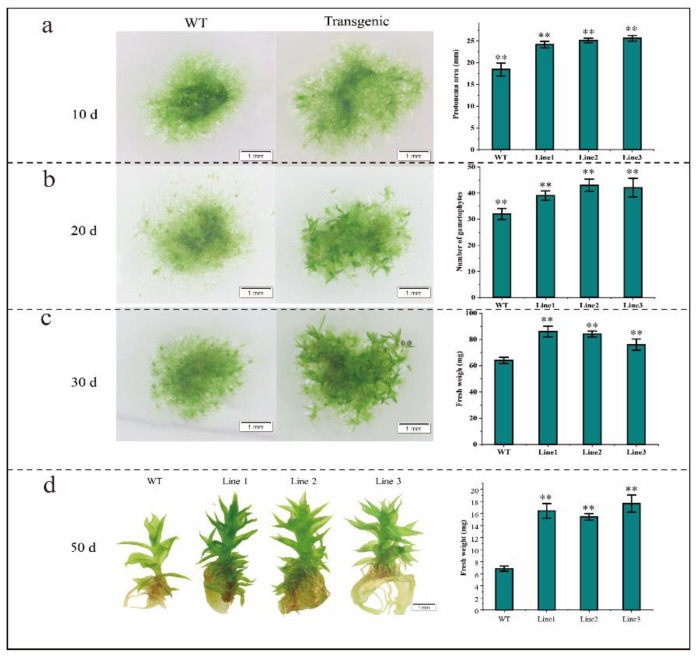
Phenotype of wild-type and *BaZFP1*-overexpressing lines of phenotype and data statistics in *P. patens*. (**a**) The growth phenotype of *P. patens* protonema transferred to KNOP medium for growth for 10 days, protonema area statistics; (**b**) the phenotype growth for 20 days, number of gametophytes statistics; (**c**) the phenotype growth for 30 days, length of gametophyte statistics; (**d**) the *P. patens* protonema were transferred to KNOP medium for growth for 50 days, phenotype and length statistics of gametophytes. Three independent biological replicates were performed, and vertical bars refer to ±SD (n = 3). Asterisks indicate significant differences between the WT and the individual transgenic lines (** *p* < 0.01).

**Figure 6 ijms-23-12894-f006:**
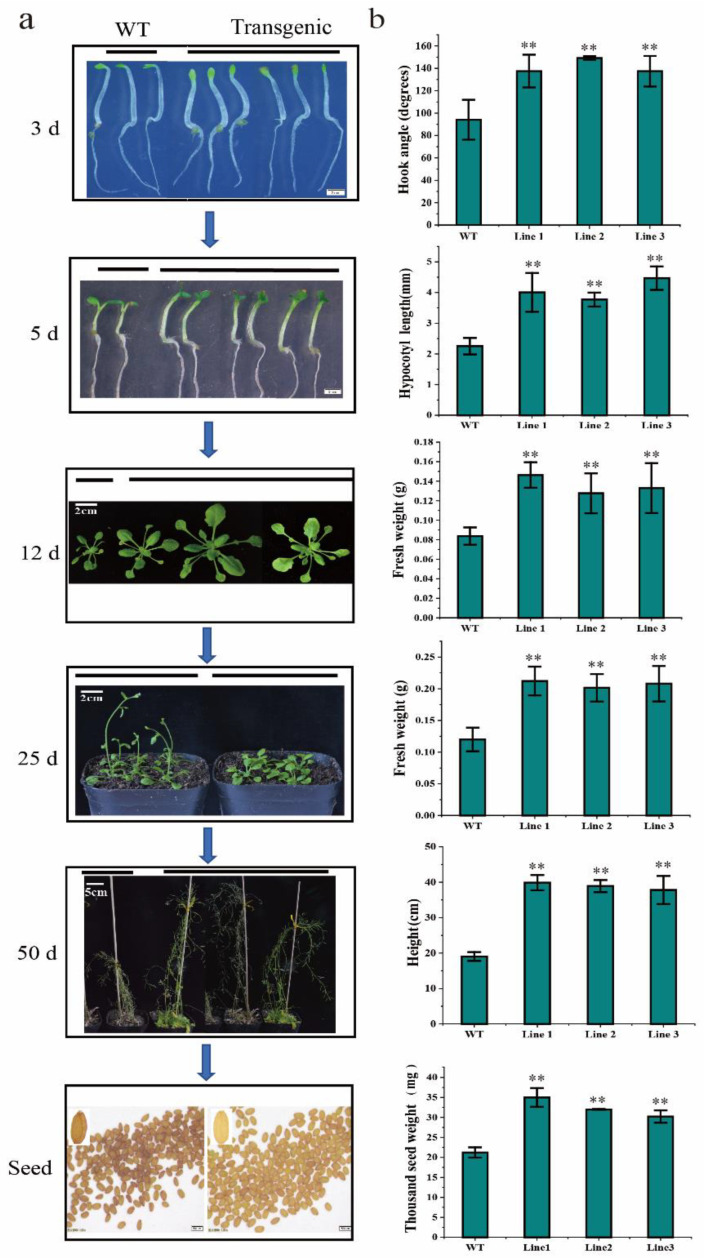
*BaZFP1* positively regulates *Arabidopsis* growth and development. (**a**) Gross morphology of the wild type and transgenic lines at different developmental stages. (**b**) Statistical analysis of thousand-seed weight, hook angle, hypocotyl length, fresh weight, and plant height at different growth stages. Seeding stage: growth in MS medium for 3 and 5 days. Growing stage: growth in nutrient soil for 12, 25, and 50 days. Three independent biological replicates were performed, and vertical bars refer to ±SD (n = 3). Asterisks indicate significant differences between WT and transgenic lines (** *p* < 0.01).

**Figure 7 ijms-23-12894-f007:**
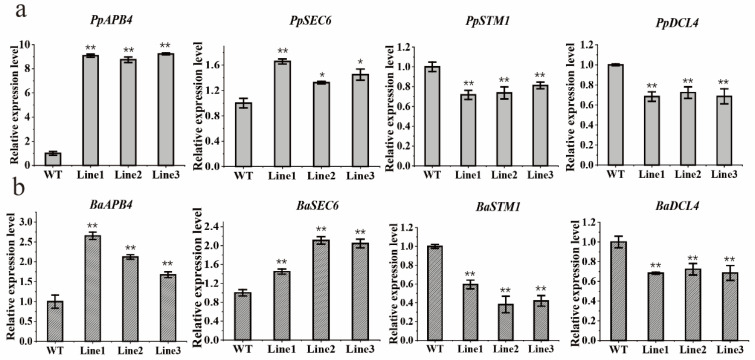
Relative expression levels of the growth- or development-related genes of *BaZFP1* in mosses. (**a**) *BaZFP1*-overexpressing *P. patens*; (**b**) *BaZFP1*-overexpressing *B. argenteum*. WT, wild type. L1–L3 are independent transgenic lines. The data are the means ± standard deviations. Asterisks indicate significant differences between WT and transgenic lines (* *p* < 0.05; ** *p* < 0.01).

**Figure 8 ijms-23-12894-f008:**
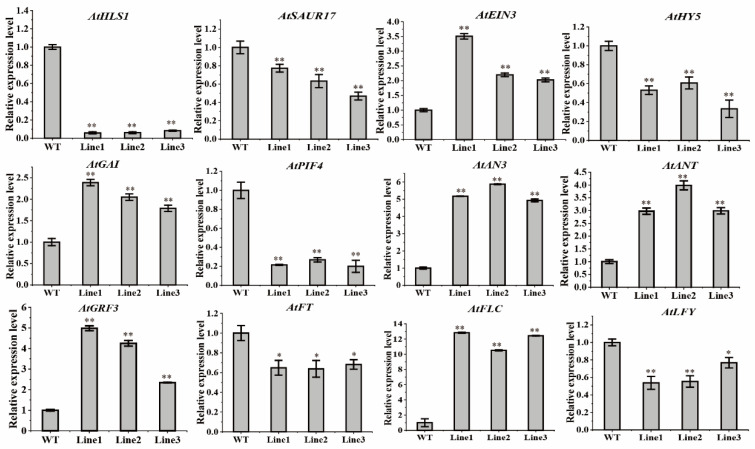
Relative expression levels of the downstream genes of *BaZFP1* in *Arabidopsis*. WT, wild type. Lines 1–3 are independent transgenic lines. The data are the means ± standard deviations. Asterisks indicate significant differences between WT and transgenic lines (* *p* < 0.05; ** *p* < 0.01).

**Figure 9 ijms-23-12894-f009:**
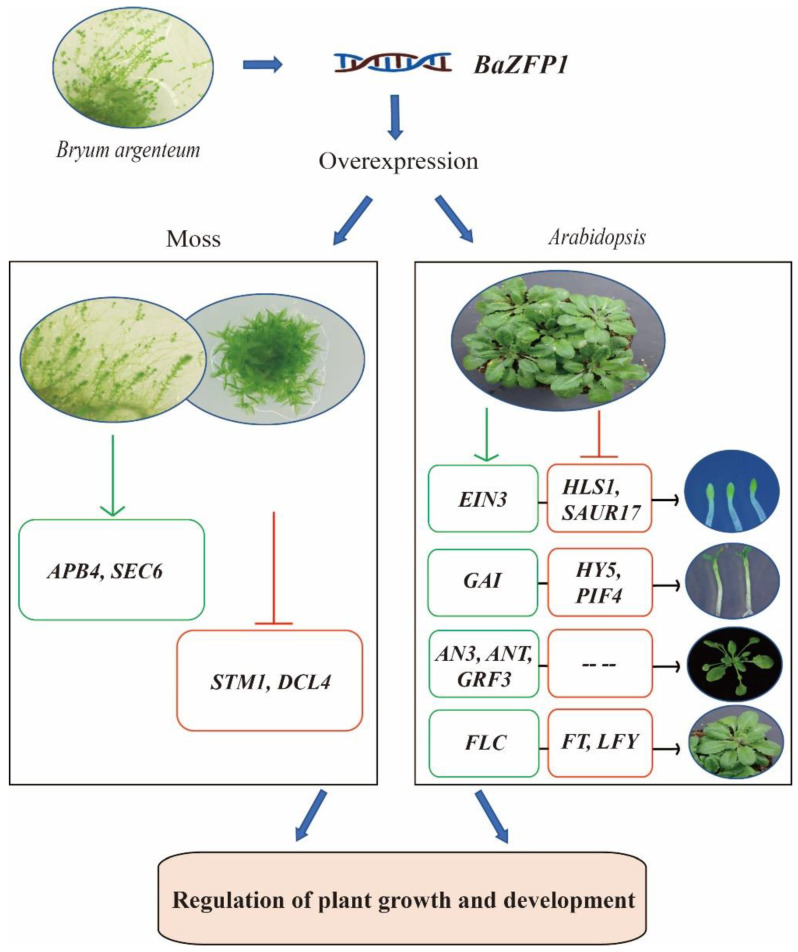
Proposed model explaining regulation of growth and plant development through *BaZFP1* overexpression. Green and red indicate a relative increase and decrease in expression level, respectively.

## Data Availability

No data were reported in this study.

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
