# Peer review of "BaZFP1,* a C2H2 Subfamily Gene in Desiccation-Tolerant Moss *Bryum argenteum*, Positively Regulates Growth and Development in *Arabidopsis* and Mosses"

_ijms, 2022, doi:10.3390/ijms232112894_

Round 1

Reviewer 1 Report

1.     Key results:

The current study identified stress-induced BaZFP1 from the desiccation-tolerance moss B. argenteum and demonstrated that BaZFP1 not only improved growth development but also promoted the differentiation process of both P. Patent and B. argenteum. The research demonstrate that BaZFP1regulates plant growth and development by regulating key growth or development-related genes in plants. This study also revealed that the C2H2 transcription factor from bryophytes plays a key role in regulating the growth and development of bryophytes and dicotyledons. It has found valuable candidates for crop breeding.

2.     Validity: The manuscript does not have a serious flaw, the abstract is clear; accessible however, there is no mention of a proposed model explaining how to use the obtained results for crop breeding and regulation of growth and plant development through BaZFP1 overexpression in crop plants. However, BaZFP1 indeed is worthy of further study.

3.     Originality and significance: The conclusions are original, but needs extra support from some updated relevant references.

4.     Data & methodology: The approach is valid, and the quality of data is fine, however, the quality of presentation must be enhanced.

5.     Appropriate use of statistics: The used statistical methods are appropriate.

6.     Conclusions: The conclusions and data interpretation are valid, but needs extra explaining of the importance of using BaZFP1 overexpression in crop plants.

7.            References: Do not contain inappropriate self-citations by authors.

Author Response

Response to Reviewer 1Comments

Point 1: The current study identified stress-induced BaZFP1 from the desiccation-tolerance moss B. argenteum and demonstrated that BaZFP1 not only improved growth development but also promoted the differentiation process of both P. Patent and B. argenteum. The research demonstrates that BaZFP1regulates plant growth and development by regulating key growth or development-related genes in plants. This study also revealed that the C2H2 transcription factor from bryophytes plays a key role in regulating the growth and development of bryophytes and dicotyledons. It has found valuable candidates for crop breeding.

Response 1: Thanks to the reviewers for your careful reading and positive evaluation of the manuscript, we will apply the BaZFP1 in crop breeding.

Point 2: The manuscript does not have a serious flaw, the abstract is clear; accessible however, there is no mention of a proposed model explaining how to use the obtained results for crop breeding and regulation of growth and plant development through BaZFP1 overexpression in crop plants. However, BaZFP1 indeed is worthy of further study

Response 2: The last paragraph in section 3.3 has added the following sentence: Using the properties that increase biomass growth and delays flowering time, the BaZFP1 can be transferred into grasses like alfalfa and some green vegetables. To explain how the BaZFP1 can be used for crop breeding.

Point 3: Originality and significance: The conclusions are original, but need extra support from some updated relevant references

Response 3: Thank you so much for your careful check. The following three relevant references are inserted where appropriate in the manuscript.

(Jiao, et al. 2020)

(Xu, et al. 2022)

(Zhu, et al. 2021)

(Liu, et al. 2022)

Jiao, Z., et al.

            2020  Genome-wide study of C2H2 zinc finger gene family in Medicago truncatula. BMC Plant Biol 20(1):401.

Liu, Y., A. R. Khan, and Y. Gan

            2022  C2H2 Zinc Finger Proteins Response to Abiotic Stress in Plants. Int J Mol Sci 23(5).

Xu, W., et al.

            2022  SMALL REPRODUCTIVE ORGANS, a SUPERMAN-like transcription factor, regulates stamen and pistil growth in rice. New Phytol 233(4):1701-1718.

Zhu, P., C. Lister, and C. Dean

            2021  Cold-induced Arabidopsis FRIGIDA nuclear condensates for FLC repression. Nature 599(7886):657-661.

Point 4:  Data & methodology: The approach is valid, and the quality of data is fine, however, the quality of presentation must be enhanced

Response 4: Thanks to the reviewers for their sincere comments, our manuscript has been submitted to the Elsevier Language Editing Services, and the corresponding certificate is the following: 

Point 5:   Appropriate use of statistics: The used statistical methods are appropriate

Response 5: Thank you to the reviewers for their positive evaluation of this manuscript

Point 6: Conclusions: The conclusions and data interpretation are valid, but needs extra explaining of the importance of using BaZFP1 overexpression in crop plants

Response 6: The last paragraph in the section conclusions has added the following sentence: Overexpression of the BaZFP1 in some grasses or green vegetables can increase biomass and delay flowering time, and greatly increase yield. To explaining of the importance of using BaZFP1 overexpression in crop plants.

Point 7: References: Do not contain inappropriate self-citations by authors.

Response 7: We appreciate the reviewer’s feedback, and we respectfully agree.

Reviewer 2 Report

Zhou et al., have utilized bio/genetic/chemical/ in silico approaches to identify the role of BaZFP1 (a C2H2 subfamily gene) in growth and development of Arabidopsis and mosses. The study is intelligible and well thought. Though, I recommend it for publication authors need to improve their english and resolution of all figures (be it supple or main).

Author Response

Response to Reviewer 2Comments

Point 1: Zhou et al., have utilized bio/genetic/chemical/ in silico approaches to identify the role of BaZFP1 (a C2H2 subfamily gene) in growth and development of Arabidopsis and mosses. The study is intelligible and well thought. Though, I recommend it for publication authors need to improve their english and resolution of all figures (be it supple or main).

Response 1: First, thanks to the reviewers for their valuable time on this manuscript and also for their valuable comments. Questions about English writing have been edited through Elsevier Language Editing Services, the corresponding certificate is the following:

On the question of image resolution, first of all, we have improved the resolution of each image to at least 300 dpi. In addition, images may be compressed to a certain extent in Word documents. We will also provide high-resolution images in the compression package.
